# Enhancing Anticoagulation Monitoring and Therapy in Patients Undergoing Microvascular Reconstruction in Maxillofacial Surgery: A Prospective Observational Trial

**DOI:** 10.3390/jpm12081229

**Published:** 2022-07-27

**Authors:** Tom A. Schröder, Henry Leonhardt, Dominik Haim, Christian Bräuer, Kiriaki K. Papadopoulos, Oliver Vicent, Andreas Güldner, Martin Mirus, Jürgen Schmidt, Hanns C. Held, Oliver Tiebel, Thomas Birkner, Jan Beyer-Westendorf, Günter Lauer, Peter M. Spieth, Thea Koch, Lars Heubner

**Affiliations:** 1Department of Oral and Maxillofacial Surgery, University Hospital Carl Gustav Carus, Technische Universität Dresden, Fetscherstraße 74, 01307 Dresden, Germany; tom.schroeder@uniklinikum-dresden.de (T.A.S.); henry.leonhardt@uniklinikum-dresden.de (H.L.); dominik.haim@uniklinikum-dresden.de (D.H.); christian.braeuer@uniklinikum-dresden.de (C.B.); kiriakikaterina.papadopoulos@uniklinikum-dresden.de (K.K.P.); guenter.lauer@uniklinikum-dresden.de (G.L.); 2Department of Anesthesiology and Intensive Care Medicine, University Hospital Carl Gustav Carus, Technische Universität Dresden, Fetscherstraße 74, 01307 Dresden, Germany; oliver.vicent@ukdd.de (O.V.); andreas.gueldner@uniklinikum-dresden.de (A.G.); martin.mirus@uniklinikum-dresden.de (M.M.); juergen.schmidt@uniklinikum-dresden.de (J.S.); peter.spieth@ukdd.de (P.M.S.); thea.koch@ukdd.de (T.K.); 3Department of General, Thoracic and Vascular Surgery, Carl Gustav Carus Faculty of Medicine, University Hospital Carl Gustav Carus, Technische Universität Dresden, Fetscherstraße 74, 01307 Dresden, Germany; hanns-christoph.held@ukdd.de; 4Institute of Clinical Chemistry, University Hospital Carl Gustav Carus, Technische Universität Dresden, Fetscherstraße 74, 01307 Dresden, Germany; oliver.tiebel@uniklinikum-dresden.de; 5Center for Evidence-Based Healthcare (ZEGV), Carl Gustav Carus Faculty of Medicine, University Hospital Carl Gustav Carus, Technische Universität Dresden, Fetscherstraße 74, 01307 Dresden, Germany; thomas.birkner@ukdd.de; 6Division of Hematology and Hemostasis, Department of Medicine I, Thrombosis Research University Hospital “Carl Gustav Carus”, Technische Universität Dresden, Fetscherstraße 74, 01307 Dresden, Germany; jan.beyer@uniklinikum-dresden.de

**Keywords:** coagulation monitoring, viscoelastic testing, maxillofacial surgery, free flap thrombosis, point-of-care coagulation

## Abstract

Background: In reconstructive surgery, loss of a microvascular free flap due to perfusion disorders, especially thrombosis, is a serious complication. In recent years, viscoelastic testing (VET) has become increasingly important in point-of-care (POC) anticoagulation monitoring. This paper describes a protocol for enhanced anticoagulation monitoring during maxillofacial flap surgery. Objective: The aim of the study will be to evaluate, in a controlled setting, the predictive value of POC devices for the type of flap perfusion disorders due to thrombosis or bleeding. VET, Platelet monitoring (PM) and standard laboratory tests (SLT) are comparatively examined. Methods/Design: This study is an investigator-initiated prospective trial in 100 patients undergoing maxillofacial surgery. Patients who undergo reconstructive surgery using microvascular-free flaps will be consecutively enrolled in the study. All patients provide blood samples for VET, PM and SLT at defined time points. The primary outcome is defined as free flap loss during the hospital stay. Statistical analyses will be performed using t-tests, including the Bonferroni adjustment for multiple comparisons. Discussion: This study will help clarify whether VET can improve individualized patient care in reconstruction surgery. A better understanding of coagulation in relation to flap perfusion disorders may allow real-time adaption of antithrombotic strategies and potentially prevent flap complications.

## 1. Introduction

Flap surgery is a key element in any reconstructive surgery. The success of flap surgery depends to a large extent on flawless blood circulation and wound healing. In oral and maxillofacial surgery, reconstructive flap surgery is mainly used in cancer therapy. 

Oral cavity carcinoma is a major cause of morbidity and mortality in patients with head and neck tumors and accounts for 1.8% of all cancer deaths worldwide [1]. 

Primary surgical resection is the therapy of choice. Surgery is mostly combined with postoperative radiation therapy or radiation combined with chemotherapy, which improves the survival rates significantly [1]. The main objective is to realize a maximized oncologic control besides a minimized impact on form and function, which is difficult because the oral cavity has a complex functional anatomy.

In order to improve the functional outcome, microvascular-free flaps are the technique of choice [2,3]. The kind of flap used depends on the tissue origin and size of the defect after tumor resection. In case of soft tissue defects, the free radial forearm flap shows excellent functional results. After segmental mandibulectomy, a fibula-free flap is used mainly for reconstruction. Besides these two frequently used free flaps, there are also others with good results.

The risk of a total flap loss is approximately 6% [4,5,6]. Although the probability of total flap loss is moderate, the consequences are tremendous as there is the risk of dramatic facial disfigurement, e.g., with total loss of the lower face. If a flap is in danger of being lost, a quick re-exploration is necessary. However, each additional flap surgery after a total flap loss increases anatomical and functional restrictions and causes additional donor site morbidity, and increases morbidity and mortality due to reoperation and anesthesia itself [7,8].

Causes of flap loss include arterial or venous occlusion due to thrombosis, compression or vessel kinking. Perfusion failure can also be caused by hematoma or edema [9]. Of the causes mentioned, thrombosis is the main reason for flap failure [5,10]. Thrombosis seems to occur more frequently when radiation has already been administered prior to surgery [11,12]. Interestingly, hypercoagulability due to malignancies, hereditary conditions and acquired thrombophilia does not seem to increase the probability of flap thrombosis [13]. In order to prevent thrombosis, adequate thromboprophylaxis should be used, but, in practice, it is difficult to determine the most effective antithrombotic regimen [13]. 

Techniques and therapies aiming at the improvement of wound healing and tissue regeneration following free flap surgery are of particular interest. As wound healing is strongly associated with optimal tissue perfusion, monitoring of anticoagulation to prevent thrombosis and bleeding is crucial. Although severe bleeding complications due to anticoagulation after maxillofacial flap surgery are rare (1.4–2.9%) [5,14], they can lead to severe airway obstructions followed by respiratory failure and hypoxemia. In postoperative hematoma, an identifiable source of bleeding is found in about 25% of cases, and flap vessel compromise occurs in 75% of cases [15]. The optimal balance between preventing thrombosis of vulnerable micro anastomosis and avoiding bleeding complications is a highly challenging task for surgeons and intensive care therapists. 

The optimal balance of anticoagulation management can be determined using various blood testing methods. Standard laboratory tests (SLT) are frequently used for anticoagulant drug adjustment. For the latter, activated partial clotting time (aPTT) is mainly used for unfractionated heparin (UFH) monitoring and prothrombin time (PT) to guide the dosing of vitamin K antagonists. Low molecular weight heparin and newer anticoagulant drugs (including oral factor Xa and thrombin inhibitors or parenteral fondaparinux and argatroban) are far more difficult to monitor because of their in vitro effect on SLT does not correlate to the in-vivo anticoagulant activity. For prevention of arterial or venous thromboembolic complications (ATE/VTE), reliable and fast identification of states of hypercoagulation and impaired fibrinolysis is crucial to determine optimal anticoagulation management [16,17,18,19]. For instance, fibrinolysis shutdown following trauma has been recognized as a major contributor to ATE/VTE in high-risk patients [18]. However, fibrinolysis capacity is not reflected by SLT or other routine assays such as d-dimer testing. Furthermore, there is no routine SLT available for platelet function monitoring [20], and aPTT and PT do not capture the effects of platelets and endothelial cells on the coagulation cascade [21].

The most important drawback of SLT is the limited value for predicting bleeding and thromboembolic complications. Whole blood test assays, often available as point-of-care (POC) devices, may overcome this disadvantage [22]. They capture all components of coagulation based on the cellular model and could potentially improve decision-making [22,23]. For instance, POC devices for viscoelastic testing (VET) can detect hypercoagulation and impaired fibrinolysis by evaluating the mechanical properties of clot formation and lysis [22,24,25].

Recent studies showed that VET was able to predict flap thrombosis in patients undergoing maxillofacial surgery [26,27]. However, these studies focused only on hypercoagulability assessed with VET. None of these studies investigated anticoagulation monitoring. Furthermore, the role of fibrinolytic capacity, which seems to have a high impact on ATE/VTE, has never been investigated in these patients before. Recent improvements in the ClotPro^®^ VET device enable physicians to assess fibrinolytic capacity using recombinant tissue plasminogen activator as a potent plasmin activator (TPA-test) [28]. Additionally, monitoring of lower molecular weight heparin is possible with the addition of snake venom RVV-V (Russell viper venom factor V activator). The RVV assay encompasses a factor X activator derived from the snake venom “Russell’s viper”. The presence of factor Xa inhibitors prolongs coagulation time in the RVV test [29]. Additionally, the adequate response to platelet inhibition drugs was investigated multiple times in patients after coronary artery intervention for ischemic coronary heart disease [30]. 

This study aims to investigate two different aspects of post-surgery care to prevent thromboses and bleeding complications: 

(1)Identifying patients at high risk for ATE/VTE using VET to detect hypercoagulability and/or impaired fibrinolytic capacity before and after surgery (Powered analysis);(2)Exploratory evaluation of POC testing for the management of anticoagulant and—if applicable—antiplatelet treatment regimens (Pilot study).

## 2. Methods: Study Setting, Participants and Outcome

### 2.1. Study Setting

This is an investigator-initiated prospective trial. It will be conducted at the Department of Maxillofacial Surgery and the intensive care units at the University Hospital Carl Gustav Carus in Dresden, Germany. Patients undergoing maxillofacial surgery with free flap reconstruction will be enrolled consecutively. Patients with indications for single platelet inhibition before surgery will be analyzed as a subgroup.

### 2.2. Eligibility Criteria

Adult patients, 18 years of age or above, who are undergoing maxillofacial surgery and receive a microsurgical flap treatment. Specifically, the following cases are selected:Patients who receive microvascular reconstruction after ablative tumor surgery;Patients who receive microvascular reconstruction for secondary reconstruction;Patients who receive microvascular reconstruction in the context of bone necrosis or osteomyelitis.

### 2.3. Exclusion Criteria

Patients who receive therapeutic anticoagulation before surgery;Patients with hereditary coagulopathies (von Willebrand disease, hemophilia A, B and C, factor VII deficiency, congenital thrombocytopenia);Patients who do not give their consent to participate in this study;Patients who receive dual platelet inhibition.

### 2.4. Test Methods

At defined time points, blood samples for laboratory analysis, POC blood gas analysis, Multiplate^®^ (Roche, Basel, Switzerland) and VET will be drawn from each patient (Figure 1).

Time points are (1) baseline measurement at the start of surgery, (2) directly before microanastomosis (MA) during surgery, (3) 1. postsurgical day (POD), (4) 2. POD, (5) 3. POD, (6) 5. POD, (7) 7. POD and (8.) 10. POD. Changes between laboratory parameters and results of VET will be analyzed for time points (2)–(8) compared to the baseline measurement. Multiplate^®^ (Roche, Basel, Switzerland) analysis will be performed only at baseline and 2nd postsurgical day to assess sufficient platelet inhibition. Additionally, one blood sample (2+) will be drawn directly from the anastomosis vein during surgery for observational analysis of local blood gas analysis.

### 2.5. Viscoelastic Testing (VET)

VET samples will be processed using arterial blood sampling. ClotPro^®^ is a recently developed VET system that uses a cup and a pin to measure clot formation, with the cup rotating via an elastic element and the pin functioning as a stationary counterpart [31,32]. The original technique was described by Hartert in 1951 [33]. In principle, the mechanical deceleration of the cup rotation is detected and translated into a viscoelastometric amplitude. ClotPro^®^ (Haemonetics, Boston, MA, USA) is available at the bedside and predominantly used in the intensive care unit (ICU), operating room or emergency department, with 9 different test kits being available for several measurements depending on the added reagent. Each assay’s specific pipette tip is loaded with the respective dried reagent. For the present study, extrinsically activated (EX) (assessment of extrinsic coagulation pathway), fibrin clot polymerization (FIB) (examination of fibrinogen level and fibrin integration), intrinsically activated (IN) (assessment of intrinsic coagulation pathway), TPA (detection of hypofibrinolysis) and RVV (detection of Xa-inhibitors) test will be used. EX and FIB tests are heparin-insensitive because hexadimethrine bromide is added, CaCI2 recalcifies the sample in both assays and recombinant tissue factor (rTF) starts the coagulation. FIB-test analyses clot formation after platelet inhibition by addition of cytochalasin D and a synthetic GP2b3a antagonist. IN-test also uses CaCI2 to recalcify the samples but also involves ellagic acid for coagulation activation. RVV test with the addition of snake venom RVV-V (Russell viper venom factor V activator) provides a possibility for rapid detection of Xa inhibitors as well as low molecular weight heparin, potentially enabling fast decision taking regarding relevant plasma levels [34]. In the TPA essay, recombinant tissue plasminogen activator (r-tPA; 650 ng/mL) is used to cleave plasminogen to plasmin as a potent activator of fibrinolysis. ClotPro^®^ (Haemonetics, Boston, MA, USA) system provides various parameters, which are visualized in Figure 2.

### 2.6. Platelet Monitoring (PM)

Multiplate^®^ analyzer (Roche, Basel, Switzerland) is the most frequently used POC device to analyze platelet function. Multiplate^®^ analyzer (Roche, Basel, Switzerland) uses different reagents for the anticipated receptor antagonists, e.g., adenosine diphosphate (ADP), collagen (COL), thrombin receptor-activating peptide (TRAP), arachidonic acid (ASPI) and ristocetin (Risto). An increase in impedance as a result of platelet aggregation is measured over time (30–40 min) and converted into aggregation units (AU) plotted against time (AU * min) [35].

### 2.7. Standard Laboratory Tests (SLT)

Standard laboratory analyses include relative prothrombin time (PT in% of normal and INR), activated partial thromboplastin time (aPTT), fibrinogen, fibrin monomers, D-dimers and Protein C/S on STA R Max3-Analyzers (STAGO Deutschland GmbH, Düsseldorf, Germany). Additional blood count analyses are performed using EDTA tubes for hemoglobin concentration, white blood cell count and platelet count. A serum collecting tube is used for measurements of inflammatory parameters (C-reactive protein (CRP) and procalcitonin (PCT)) and organ function monitoring (creatinine, bilirubin and albumin). The activity of anti-Xa inhibition (LWMH) in a patient’s plasma is measured using the one-stage chromogenic assay STA^®^-Liquid Anti-Xa assay (Stago, Asnières-sur-Seine, France). Chromogenic assay analyzes the substance of residual factor Xa of the patient’s plasma to hydrolyze a chromogenic substrate by measuring the change in optical density per minute, which is inversely correlated with the concentration of Xa inhibitor (LWMH) in the patient’s plasma. Blood gas analysis will be performed at the bedside using ABL Flex90 systems (Radiometer, Brønshøj, Denmark).

### 2.8. Anticoagulation Management

Every patient receives standard thrombosis prophylaxis with 4000 international units (IU) of low molecular weight heparin (enoxaparin sodium) once daily. If patients weigh over 90 kg, the thrombosis prophylaxis is adjusted to 6000 IU and over 120 kg to 8000 IU (divided into two doses). Thrombosis prophylaxis is also adapted to kidney function. If the glomerular filtration rate (GFR) is between 15 and 30 mL/min/1.73 m^2^, the dose is reduced to 2000 IU. If GFR is below 15 mL/min/1.73 m^2^, thrombosis prophylaxis will be changed to unfractionated heparin (heparin natrium) 5000 IU twice a day.

Intraoperative adjustment of the predefined anticoagulation management is based on the surgeon’s observations during microsurgery. In case of arterial plaques or the observation of intravascular thrombosis during the reopening of the microanastomosis, single platelet inhibition will be additionally used to prevent arterial thrombosis Microanastomosis will be re-performed if perfusion mismatch clinically occurs, which is verified by using an intraoperative Doppler device. Aspirin 100 mg given intravenously during surgery, followed by oral administration for 8 weeks, will be generally used. In the case of perioperatively administered aspirin, a Multiplate^®^ (Roche, Basel, Switzerland) analysis will be performed on the 2nd postoperative day (POD), as depicted in Figure 2. Individually increased thrombosis prophylaxis based on the surgeon’s decisions will be recorded in case report forms (CRF).

### 2.9. Assessments of Flap Complications

After the operation, clinical bedside examinations will be scheduled at standardized time points (Figure 1). Visual inspection and assessment of the capillary refill time serve as assessment criteria. If there is no skin flap, e.g., in some fibular grafts, the assessment is performed using a Doppler device. Assessment times follow a fixed protocol and are mainly recorded by specially trained nursing staff. Hourly assessment for the first 48 h, every 4 h on POD 3 and 4, every 8 h on POD 5 and 6 and interim visits at any time in case of clinical complications.

## 3. Methods: Statistical Analysis

### 3.1. Primary Outcome (Powered Analysis)

The primary outcome is the overall free flap failure and the correlation with results of VET, defined in the following primary endpoints:MCF parameter in the INtest at baseline and occurrence of free flap failure;CFT parameter in the INtest at baseline and occurrence of free flap failure;A10 parameter in the INtest at baseline and occurrence of free flap failure.

### 3.2. Sample Size

By estimating for mean differences and standard deviations for thrombosis and non-thrombosis groups in VET in the literature [26] and focusing on the goal of conducting three primary comparisons of means between the thrombosis and non-thrombosis group at alpha = 0.05/3 = 0.0166 (Bonferroni adjustment) with a power of 80%, a sample size of 100 patients will be sufficient. Means and standard deviations reported in the literature are directly transferable as assumptions for the sample size calculation for the MCF, CFT and A10 parameters of the INtest for the VET technology used in this study.

### 3.3. Statistical Analysis

Each primary endpoint is analyzed as a comparison of differences in differences. Specifically, the difference in the mean change from baseline to the time points (*t_i_*) for the free flap failure (yes) group versus the mean change from baseline to the time points (*t_i_*) for the free flap failure (no) group is tested via a *t*-test. The post-baseline time points *t_i_* to be included in the primary analysis are those at which free flap failure occurs. Patients with free flap failure will contribute their baseline value and one post-baseline value to the analysis of each primary endpoint. Patients without free flap failure will contribute their baseline value and likely more than one post-baseline value to the mean calculation for each primary endpoint (since free flap failure will likely occur/be recorded at different time points for different patients).

### 3.4. Power

The study is powered at 80% at alpha = 0.05 (two-sided) to detect a difference in the change from time point 1 to time points *t_i_* means of the free flap failure versus non-failure group for the three VET tests/parameter combinations specified as primary endpoints if such difference exists.

### 3.5. Secondary Outcomes (Exploratory Study)

Secondary outcomes are the evaluation of adequate response to anticoagulation therapy and—if applicable—to platelet inhibition drugs in POC testing as well as SLT. Furthermore, the correlation between flap failure and the occurrence of any VTE/ATE during the in-hospital stay will be analyzed to all VET and SLT parameters as a pilot study. Patients with antiplatelet therapy (APT) before surgery will be analyzed as a subgroup in an exploratory manner.

### 3.6. Statistical Analysis

The same analysis approach will be applied to the other test-parameter combinations available from the VET technology. Details will be specified in the Statistical Analysis Plan (SAP).

### 3.7. Supportive Analyses

A graphical approach will be applied to depict the patient level trajectories of each test/parameter combination over time (a) for the free flap failure (no) group and (b) from baseline to time point of free flap failure for the free flap failure (yes) group.

Mean comparisons for each test/parameter combination between the groups in change from baseline to each post-baseline time point will also be performed.

Consideration will be given to the use of thrombolytic drugs in case of pulmonary embolism. Data from patients receiving those will be excluded from this supportive analysis starting with the time point where administration began.

An extension of the Cox proportional hazards model incorporating time-varying covariates will be applied with all clinically relevant test/parameter combinations entering the model as covariates. The time-to-event outcome will be thrombosis.

### 3.8. Data Exclusion

Data will be excluded if outside the possible range for the parameter/test. Outliers will be examined but not automatically excluded.

## 4. Methods: Recruitment, Data Collection and Management

### 4.1. Recruitment

Patients who are admitted to the department of maxillofacial surgery to undergo reconstructive surgery using microvascular free flaps will be consecutively enrolled in the study. Investigators have electronic access to incoming patient data and will be able to detect candidates by the time of presentation. With an average of 90 patients admitted to our department every year with a main diagnosis of cancer in the maxillofacial area, osteomyelitis or necrosis of the lower jaw who need reconstructive surgery, an overall recruitment period of two years is anticipated, assuming that 80% of the admitted patients meet the eligibility criteria and 20% drop-out of the study. A total of 100 patients are planned to be enrolled in the study.

### 4.2. Data Collection Methods

Laboratory parameters will be obtained by technical staff at the central laboratory as part of the routine diagnostics and provided via the clinical information system. ClotPro^®^ and Multiplate^®^ assays will be run simultaneously for all participants. Data will be primarily collected and stored offline in an Excel sheet (Microsoft Corporation, Albuquerque, NM, USA).

### 4.3. Data Management

All clinical data will be transferred to a secure, password-protected online database. Backup copies will be stored offline at the trial center.

### 4.4. Data Monitoring

This is a non-interventional observational study. Measurements of the patient’s blood will be performed in vitro, and results will not be incorporated into treatment decisions. Therefore, data monitoring is not needed.

### 4.5. Harms

Given the nature of the study, no harm is expected.

### 4.6. Auditing

No auditing will be necessary.

## 5. Discussion

Free flap thrombosis is a severe complication after reconstructive surgery. Current principles of anticoagulation monitoring and therapy are far away from individually tailored care. The aim of this study is to investigate patients’ individual characteristics that lead to prothrombotic state and postsurgical complications. Potentially this approach might enable physicians to perform more personalized anticoagulation treatment, which should be analyzed in future controlled trials.

## Figures and Tables

**Figure 1 jpm-12-01229-f001:**
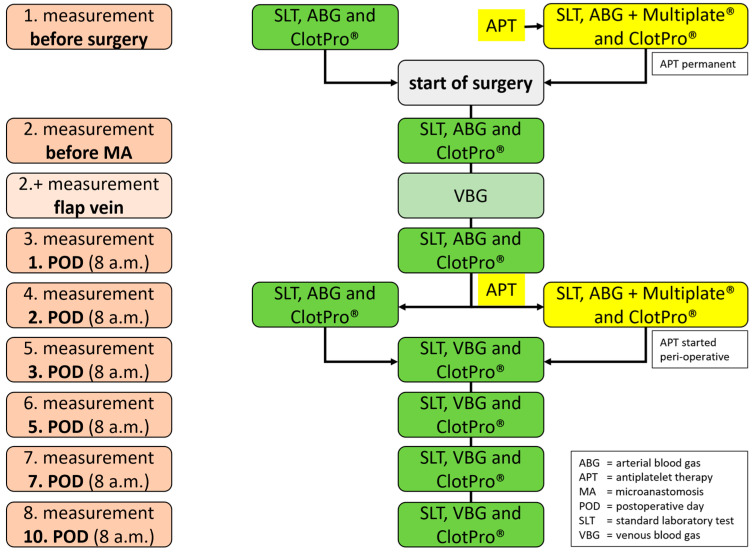
Flowchart: times and types of laboratory tests (ABG = arterial blood gas; APT = antiplatelet therapy; MA = microanastomosis; POD = postoperative day; SLT = standard laboratory test; VBG = venous blood gas; ClotPro^®^ (Haemonetics, Boston, Massachusetts, USA); Multiplate^®^ (Roche, Basel, Switzerland).

**Figure 2 jpm-12-01229-f002:**
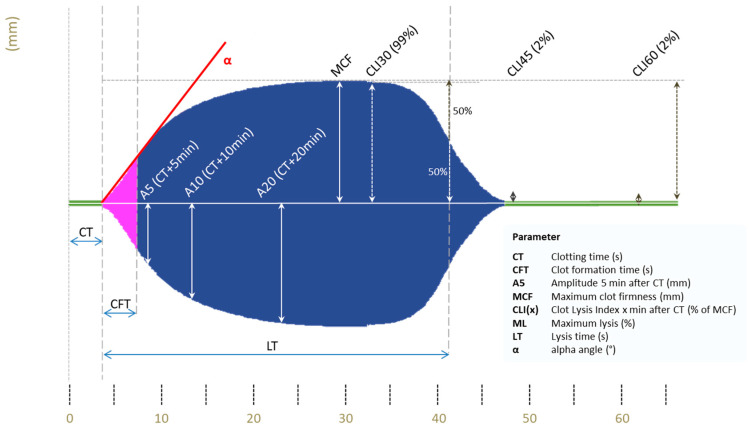
ClotPro^®^ Parameters (Permission Haemonetics^©^ (Haemonetics, Boston, MA, USA)), y-axis = distance in mm and x-axis = time in minutes, CT = Clotting time (s); CFT = Clot formation time (s); A5 = Amplitude 5 min after CT (mm); MCF = Maximum clot firmness (mm); CLI(x) = Clot Lysis index x min after CT (% of MCF); ML = Maximum lysis (%); LT = Lysis time (s); α = alpha angle (°).

## Data Availability

The final trial dataset will be available to all investigators of the study.

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
