# Peer review of "Enhancing Anticoagulation Monitoring and Therapy in Patients Undergoing Microvascular Reconstruction in Maxillofacial Surgery: A Prospective Observational Trial"

_jpm, 2022, doi:10.3390/jpm12081229_

Round 1
Reviewer 1 Report
This manuscript described their prospective study about the anticoagulation monitoring in patients undergoing microsurgery in maxillofacial reconstruction. This topic is very important in microsurgery to keep high success rate. The contents were relevant and interesting, clear and easy to read.
This topic described the importance of VET, PM and SLT monitoring in microsurgery. Its concept has originality. I suppose VET(viscoelastic testing) monitoring in microsurgery might be new concept compared with other published material. It might be preliminary report of study. Anyway, Its concept might be interesitng.
Author Response
We thank Reviewer for comments and appreciate his oppinion.
Reviewer 2 Report
Interesting and well-structured study. The outcomes might contribute to a more personalized anticoagulation treatment during perioperative period of microsurgical reconstruction.
Author Response
We appreciate Reviewers opinion and we agree in his expectation that our results might contribute to a more personalized anticoagulation treatment.
Reviewer 3 Report
Dear Author,
Thank you for the opportunity to review your submitted Study Protocol Scrip entitled “Enhancing anticoagulation monitoring and therapy in patients undergoing microvascular reconstruction in maxillofacial surgery: a prospective observational trial”.
Although the topic is not new, the issue has not lost in actuality and to date it still lacks an adequate study.
Overall, I would like to congratulate you on this very successful protocol.
My only criticism is that the adjustment of the intraoperative anticoagulation management depends on the opinion of the surgeon. Your entire coagulation monitoring is objectified by measurable data, so the opinion of the surgeon is a rather soft factor, which needs strengthening.
How do you define a complicated anastomosis?
Do you use a coppler device?
Despide, the protocol is rather coherent.
Author Response
We thank Reviewer for his comments. The term “complicated anastomosis” is misleading. Therefore, we appreciate much for pointing this out.
The authors define a “complicated anastomosis” as a recurrent perfusion mismatch after the initially sufficient microanastomosis with the absence of any surgical related error. The perfusion mismatch is objectified by using an intraoperative Doppler device. This leads to the immediate need to renew the microanastomosis, sometimes several times. If an intravascular thrombus is seen during re-anastomosis, single platelet inhibition will be used to prevent from further arterial thrombosis.
We don’t use a coupler device.
The suggestions were included in the manuscript.